METHODS

# Quantifying HiPSC-CM structural organization at scale with deep learning-enhanced SarcGraph

**Saeed Mohammadzadeh**[ID][1], **Emma Lejeune**[ID][2]*

**1** Division of Systems Engineering, Boston University, Boston, Massachusetts, United States of America,
**2** Department of Mechanical Engineering, Boston University, Boston, Massachusetts, United States of America

* elejeune@bu.edu

## Abstract

In cardiac cells, structural organization is an important indicator of cell maturity and healthy function. Healthy and mature cardiomyocytes exhibit a highly organized structure, characterized by well-aligned almost crystalline morphology with densely packed and organized sarcomeres. Immature and/or diseased cardiomyocytes typically lack this highly organized structure. Critically, human induced pluripotent stem cell-derived cardiomyocytes (hiPSC-CMs) offer a valuable model for studying human cardiac cells in a controlled, patient-specific, and minimally invasive manner. However, these cells often exhibit a disorganized and difficult to quantify structure both in their immature form and as disease models. In this work, we extend the SarcGraph computational framework—designed specifically to assess the structural and functional behavior of hiPSC-CMs—to better accommodate the structural features of immature cells. There are two key enhancements: (1) incorporating a deep learning-based z-disc classifier, and (2) introducing a novel ensemble graph-scoring approach. These modification significantly reduced false positive sarcomere detections, particularly in immature cells, and improved the detection of longer myofibrils in mature samples. With this enhanced framework, we analyze an open-source dataset published by the Allen Institute for Cell Science, where, for the first time, we are able to extract key structural features from these data using information from each individually detected sarcomere. Not only are we able to use these structural features to predict expert scores, but we are also able to use these structural features to identify bias in expert scoring and offer an alternative unsupervised learning approach based on explainable clustering. These results demonstrate the efficacy of our modified SarcGraph algorithm in extracting biologically meaningful structural features, enabling a deeper understanding of hiPSC-CM structural integrity. By making our code and tools open-source, we aim to empower the broader cardiac research community and foster further development of computational tools for cardiac tissue analysis.

**Data availability statement:** Code and information related to the models and

methodology presented here are available at https://github.com/saeedmhz/sarcgraph-image. The original SarcGraph implementation is available on GitHub at https://github.com/Sarc-Graph/sarcgraph. Supplementary data, including pretrained models and extracted features, are available at https://dataverse.harvard.edu/dataset.xhtml?persistentId=doi:10.7910/DVN/3AIJKU.

**Funding:** This study was funded by the National Science Foundation CELL-MET ERC (EEC-1647837, https://www.nsf.gov/) and the American Heart Association Career Development Award (856354, https://www.heart.org/), awarded to EL. The funders had no role in study design, data collection and analysis, decision to publish, or preparation of the manuscript. EL received salary support from both the National Science Foundation and the American Heart Association. SM received salary support from the American Heart Association.

**Competing interests:** The authors have declared that no competing interests exist.

## Author summary

Heart disease remains a leading cause of morbidity and mortality worldwide. To better understand cardiac health and advance new therapies, researchers are increasingly turning to human induced pluripotent stem cell-derived cardiomyocytes (hiPSC-CMs). These cells provide a patient-specific platform for studying heart disease and developing engineered heart tissue. However, analyzing hiPSC-CMs remains challenging because their structural organization, especially in immature or diseased states, often appears disorganized under the microscope, making it difficult to extract meaningful quantitative insights. In this study, we present significant enhancements to our computational tool, SarcGraph, which is designed to detect and analyze sarcomeres, the fundamental units responsible for muscle contraction. Our improvements allow for more accurate detection of sarcomeres, even in structurally disorganized cells. We applied the updated tool to a large open-access dataset and demonstrated that it can not only replicate expert assessments of cell structure but also uncover inconsistencies in manual scoring and provide a robust automated alternative. By releasing our methods as open-source software, we aim to support the broader cardiac research community in studying hiPSC-CM structure and in developing more consistent, scalable approaches for assessing cellular growth, maturity, and disease-related changes.

## 1. Introduction

Cardiac disease remains the leading cause of death worldwide, prompting extensive research efforts towards developing technology to repair the damaged heart [1–4]. Since the advent of induced pluripotent stem cell (iPSC) technology nearly two decades ago [5], human iPSC-derived cardiomyocytes (hiPSC-CMs) have emerged as a promising tool in cardiac disease research [6–8]. These cells have the potential to advance a variety of applications, from patient-specific drug testing [9,10], to disease modeling [11,12], to serving as the building blocks for patient-specific engineered heart tissue [13]. However, because hiPSC-CMs often exhibit a disorganized and difficult to quantify structure, both in their immature form and as disease models, it can be challenging to extract quantitative insight from hiPSC-CM based experiments [14].

To this end, there has been significant effort towards developing computational tools to assess the structural organization of hiPSC-CMs [15–17]. In particular, for images with fluorescently labeled striations (e.g., z-disc proteins), there are multiple image analysis tools designed to capture the structural organization of sarcomeres and myofibrils [18–20]. Sarcomeres, the fundamental contractile units of striated muscles, play a critical role in the contraction and relaxation of the heart [21]. And, mutations in genes related to sarcomeres have been linked to cardiac disease such as hypertrophic cardiomyopathy [22]. Thus, there have been extensive efforts to study sarcomere organization specifically [23,24]. For example, Morris et al. developed "ZlineDetection" [19], a computational tool that effectively detects z-discs in relatively mature cells with aligned structures and uses the detected z-discs to quantify z-disc alignment and sarcomere length (the distance between z-discs). Sutcliffe et al. introduced SarcOmere Texture Analysis (SOTA) [17], which uses Haralick features [25,26] to quantify similar sarcomeric properties. Pasqualini et al. [27] developed a computational framework to extract a set of 11 features from alpha-actinin channel images of cardiomyocytes, describing sarcomeric organization. Their work demonstrated a clear distinction between more mature and less mature cells, as the former exhibit well-oriented z-discs, reflected by elevated values

of features such as the orientational order parameter and sarcomeric packing density. SarcOp-tiM [20], another tool, uses Fast Fourier Transforms to track sarcomere length and is readily available as an ImageJ plugin. In addition, Pardon et al. recently published CONTRAX, a tool that measures hiPSC-CM contractile dynamics with traction force microscopy [28]. While these methods are highly effective in certain scenarios, they often provide quantities that are averaged over the entire image rather than analyzing a detected set of individual sarcomeres collectively. This limits their utility for detailed structural analyses of the sub-cellular components of hiPSC-CMs.

Recently, there have been multiple software tools developed to quantitatively analyze the structure and function of hiPSC-CMs that rely on segmenting individual sarcomeres [18,29, 30]. In 2019, Toepfer et al. introduced SarcTrack [29], a tool for segmentation and tracking of individual sarcomeres in movies of beating hiPSC-CMs that enables the study of high-level features including contraction rate and sarcomere shortening on individual sarcomere basis. Later, in 2021, our group published SarcGraph [30], a Python package for automatic detection and tracking of individual z-discs and sarcomeres in movies of hiPSC-CMs and continues to maintain and update this tool [31]. In addition to improved detection and tracking of z-discs and sarcomeres compared to other available tools, SarcGraph is able to extract more complete structural information. Specifically, by adding each sarcomere to a spatial graph, SarcGraph is capable of detecting myofibril chains, and of extracting additional high-level information about sarcomere organization and contractile behavior [31].

However, these existing methods, including SarcGraph, face limitations in reliably detecting z-discs and sarcomeres in hiPSC-CMs that exhibit lower levels of maturation. Because initial z-disc detection is imperfect, the quality of all downstream metrics and subsequent analysis suffers. In this work, we extensively modify SarcGraph to improve the detection of z-discs and sarcomeres, enabling a more robust analysis of structural features in hiPSC-CMs. Then, using an open-access dataset published by the Allen Institute for Cell Science [32], we leverage the enhanced SarcGraph framework to extract high-level structural features from individual sarcomeres detected within each cell. These features are then employed in both supervised and unsupervised models to quantify cell structural organization. The supervised model uses expert-assigned scores to evaluate structural organization, while the unsupervised approach eliminates the need for manual scoring, thus providing an alternative for assessing cellular organization.

The remainder of this paper is organized as follows. First, we introduce the dataset used in this work and highlights the challenges associated with manual scoring. Next, we provide an overview of the improved SarcGraph sarcomere detection pipeline—including the addition of a deep learning-based z-disc classifier—and outlines the development of machine learning models trained to quantify cell structural organization. We also include details on the features, models, and evaluation metrics employed in our analysis. In the Results section, we compare the performance of the modified SarcGraph pipeline against the original version, explore the limitations of manual scoring in greater depth, and provide a comprehensive evaluation of the machine learning models developed for quantifying cell structural organization. Overall, we view this work both as an important step towards more effective analysis of biomedical imaging data at scale, and as a showcase of significant improvements made to a computational tool designed to benefit the cardiac research community.

## 2. Dataset

The work presented in this paper is based on a publicly available dataset published by the Allen Institute for Cell Science of integrated transcriptomics and structural organization data

in hiPSC-CMs [32,33]. The dataset contains approximately 2,900 individual images, which corresponds to approximately 31,000 single cells. In the original work, the authors developed a framework to extract local and global features characterizing sarcomere organization and cell structure to quantify overall cell organization during cardiomyocyte differentiation. They introduced the Combined Organizational Score (COS), a quantitative metric that integrates these features to assess cell organization levels among individual cells. Since publication, this dataset has become a valuable resource for researchers studying cardiomyocytes development and cellular organization [34,35]. For instance, Le et al. used these hiPSC-CM images to validate the generalizability of their self-supervised deep learning model, CytoSelf, on cells with different morphology, while Kobayashi et al. leveraged the dataset to develop SarcNet, a deep learning-based framework, for predicting cellular organization scores from microscopy images.

The dataset includes both live and fixed cell images:

- **Live cells** were imaged using brightfield, Nuclear Violet LCS1 for nuclear staining, GFP-tagged alpha-actinin-2 to visualize sarcomeres, and a plasma membrane marker to delineate cell boundaries. Automated segmentation was performed using CellProfiler [36], identifying approximately 18,000 cells from 1,700 fields of view (FOVs).
- **Fixed cells** were imaged after RNA fluorescence in situ hybridization (RNA FISH) to assess transcript abundance. Imaging channels for these cells include brightfield, DAPI for nuclear staining, GFP-tagged alpha-actinin-2, and two channels to capture transcript abundance. Manual segmentation was applied, identifying approximately 13,000 cells from 1,200 FOVs.

Imaging was conducted at three distinct time points: Day 18, Day 25, and Day 32, to capture the progression of cardiomyocyte maturation. The dataset also includes metadata such as cell age (time point of imaging), cell area, and sarcomere organization features. These features include local patterns (e.g., ratios of pixels classified as background, fibers, disorganized puncta, and organized z-discs through deep learning-based semantic segmentation) and global metrics derived from Haralick correlation plots, such as peak distance and peak height. Additionally, expert scores quantifying sarcomere organization are assigned to a subset of approximately 6,000 fixed cells and 1,000 live cells. We refer readers to the original publication for details of both the experimental and data curation protocol [32].

In our analysis, we focus exclusively on the images of cells with expert-annotated organization scores (approximately 7,000 cell images in total), using the alpha-actinin-2 channel to study the relationship between sarcomere features and overall cell organization. Alpha-actinin-2, a critical protein in sarcomeres, initially appears as fibrous structures or puncta-like z-bodies in immature cells and progressively organizes into structured z-discs as the cells mature [27,37]. Fig 1 panel A-ii, shows a single cell segmented from a larger field-of-view image of multiple cells in the alpha-actinin-2 channel (A-i). In panel (A-iii), we see that potential z-disc structures have been visibly delineated. Later in the Methods section, we explain how we use SarcGraph to extract sarcomere-related features from the images of segmented single cells and explore their relationship with cell structural organization.

## 2.1. Note on manual scoring

In conjunction with the imaging data described above, the authors also performed manual scoring on a subset of the cells. Specifically, two experts manually scored approximately 5,000 RNA FISH cells, with scores ranging from 1 (indicating less organized, sparse cells with

mainly z-bodies visible) to 5 (indicating highly organized cells with primarily z-discs aligned along one axis). Representative examples of each score are shown in Fig 1 panel B. In addition to the original 5,000 scored cells, expert manual scoring was also performed on 1,000 RNA FISH cells from a different experiment, and 1,000 live samples. In the original investigation [32], the 5,000 example RNA FISH group was treated as the "training" dataset for predicting cell organization scores from image features, whereas the 1,000 RNA FISH group and the 1,000 live sample group were treated as the "test" datasets for exploring score prediction on unseen data.

Manual score labeling provides us with a unique twofold opportunity. First, we are able to explore the efficacy of our computational framework (Sarc Graph) at extracting features from images for the purpose of predicting manual scores. Second, we are able to use these datasets to quantitatively investigate the efficacy of manual scoring as a research tool. Overall, we commend the original authors for their substantial effort in creating and manually scoring this dataset, a task that is both time-consuming and complex. Here, we also briefly recognize the inherent limitations of manual scoring methods. Specifically, we highlight three key known limitations of manual scoring:

- **Limitations of Ordinal Scales**: Ordinal scales (the basis of manual scoring), where data is categorized and ranked but the relationship between ranks is not quantitatively defined, have inherent limitations. First, ordinal scales lack the resolution to differentiate subtle variations within a category and can thus mask significant differences between samples. Moreover, ordinal scales exhibit non-equidistant properties, where the intervals between adjacent scores are not uniform. This can lead to distortions, such as a score of 1 and 2 not representing the same degree of difference as a score of 2 and 3.
- **Human Rater Biases**: Numerous studies have documented biases in human ratings, such as Central Tendency Bias [38] and Recency Bias [39]. These biases can inadvertently influence the scoring process.
- **Inter-Rater Discrepancy**: As shown in panel C of Fig 1, manual scoring, even by experts, is susceptible to variability. In this dataset, we observed moderate agreement between the raters, with an intraclass correlation coefficient (ICC3) of 0.68 [40], which falls into the moderate reliability category according to the guidelines proposed by [41]. This level of agreement is consistent with a Pearson correlation coefficient of 0.65 and a simple agreement rate of 0.66, which measures the proportion of cases where raters exactly agree. These findings highlight the subjective nature of the ratings.

In light of these challenges, we aim to explore methods that could potentially address the limitations of manual scoring, leveraging the strengths of the existing dataset and considering more objective and scalable approaches to assess cell organization. In the Methods section, we detail our methodology for applying SarcGraph to detect sarcomeres and extract sarcomere organization-related features to quantify the structural organization level of individual cells.

## 3. Methods

In this work, we make significant advances to our SarcGraph software framework for quantifying the structural organization of cardiomyocytes. Specifically, the original implementation of SarcGraph does not perform well for immature cardiomyocytes that contain significant numbers of labeled components that are not z-discs (e.g., puncta), and does not perform well for single images (i.e., it relies on the presence of the multiple frames available with timelapse imaging to help filter artifacts). Thus, we introduce multiple fundamental methodological

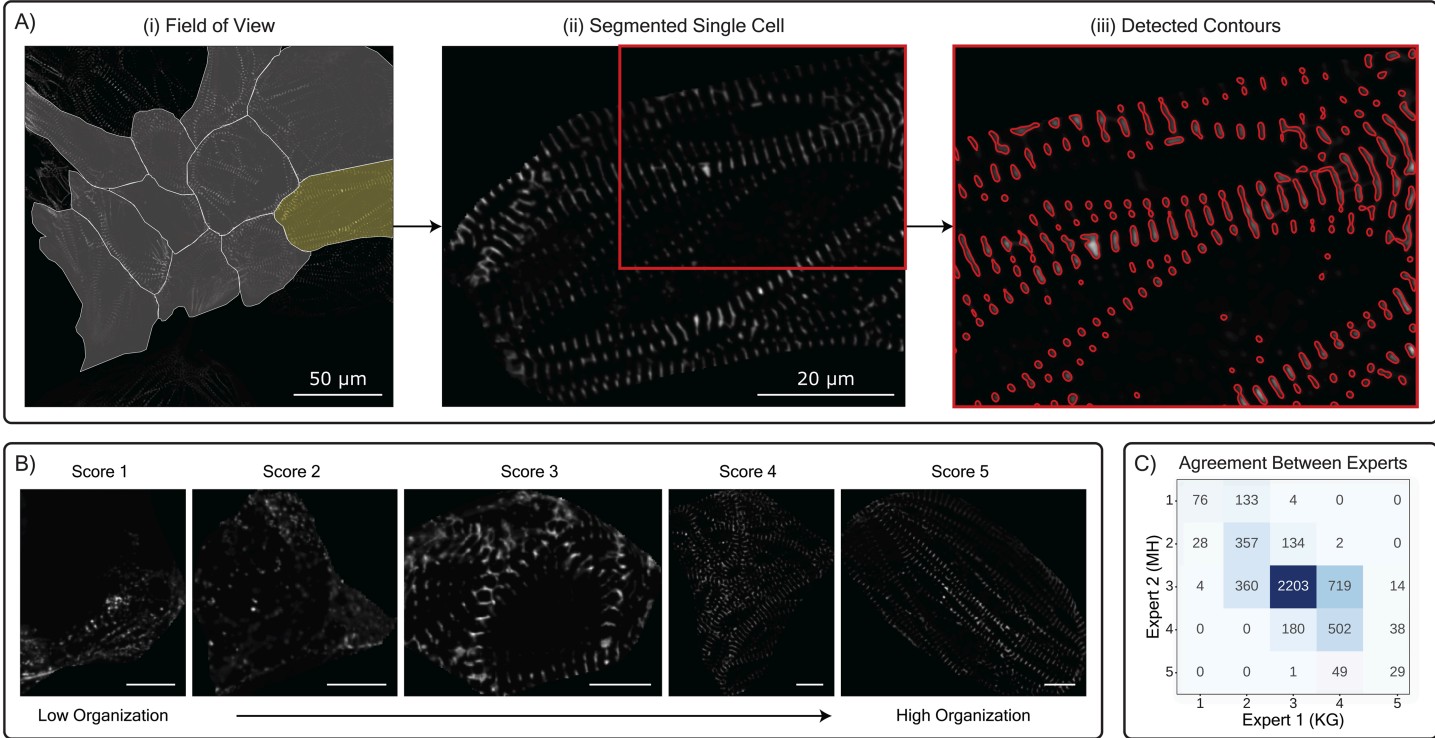

**Fig 1. Schematic representation of selected data from the Allen Institute dataset.** Note that the original dataset contains additional channels and information not depicted here. Panel (A) shows: (i) a Field of View (FoV) from a representative RNA FISH well displaying multiple cells with segmentation overlays, (ii) a single cell segmented from the field of view image (highlighted in yellow), and (iii) detected contours of potential z-discs within the selected cell, obtained by applying SarcGraph to the alpha-actinin-2 channel image. (B) Representative cells from each expert-assigned organization score group (scores 1-5). Score 1 represents poorly organized, sparse cells with predominantly z-bodies, while score 5 indicates highly organized cells with z-discs aligned along a single axis. Scale bars in individual cell images represent 10 μm. (C) Confusion matrix showing the agreement between two expert annotators in scoring cell organization.

modifications to make SarcGraph a significantly more robust tool for the research community. These improvements include both refining z-disc detection using deep learning techniques, and incorporating additional procedural steps to improve overall sarcomere detection. Following this enhanced sarcomere detection framework, we extract a suite of sarcomere-related features and use manual expert scores in conjunction with these features to develop an objective approach for quantifying cell organization. The subsequent sections detail the methodologies employed to do this.

### 3.1. Overview of the SarcGraph framework for Sarcomere detection

The SarcGraph framework for sarcomere detection consists of two key phases: z-disc segmentation and sarcomere detection. In the z-disc segmentation phase, SarcGraph first processes raw input images using a Laplacian of Gaussian (LoG) filter [42] to enhance edge features. It then applies Otsu's thresholding [43] to binarize the filtered image, enabling the detection of relevant structures. Next, the marching squares algorithm [44], from the scikit-image library [45], identifies contours, as shown in Panel (A-iii) of Fig 1. These contours are then classified as z-discs based solely on their length (contours with lengths less than 15 or above 200 pixels are filtered out)—a simplistic approach similar to the length-based filtering used to differentiate z-bodies from z-discs in related work [46]. Then, SarcGraph calculates the centroids of these z-disc contour vertices and assigns them as the z-disc locations, which are used in the sarcomere detection phase.

In the sarcomere detection phase, SarcGraph constructs a graph where the nodes of the graph correspond to detected z-discs and the edges correspond to potential sarcomeres. To build the graph, each node is first connected to its $N$ nearest neighbors, provided mutual connectivity exists between them. SarcGraph then evaluates each edge by scoring it based on its length, the angle it forms with neighboring edges, and its length consistency relative to adjacent edges. Once scored, the framework applies a rule-based pruning algorithm to eliminate edges that are less likely to represent valid sarcomeres, retaining only the most plausible candidates. The remaining edges form the final set of detected sarcomeres. For a more comprehensive explanation of the algorithm, we refer readers to our prior publication and code base [31].

## 3.2. Development of a deep learning-based Z-disc classifier

The original z-disc segmentation framework in SarcGraph has two main drawbacks: (1) the use of global Otsu thresholding, and (2) the reliance on contour length for z-disc classification. Global Otsu thresholding applies a single pixel value threshold across the entire image, which fails to account for local variations in contrast. Likewise, classifying contours based solely on length overlooks other potentially important structural features. As a result, this simplistic classification can lead to a high rate of false z-disc detection, particularly in immature cells where structures such as z-bodies or fibrous formations are more prevalent, as shown in Fig 2 panel A.

Deep learning models have demonstrated remarkable performance in handling complex image analysis tasks, especially in medical imaging applications, including cardiac tissue analysis [47–51]. For instance, Neininger et al. [46] developed SarcApp, which uses a U-Net [52] to improve image binarization, replacing traditional methods like Otsu thresholding; however, their approach still relies on length-based thresholds for z-disc classification. To address this challenge, we propose a deep learning-based approach to classify z-discs by allowing the model to learn discriminative features from contour images without explicitly specifying them. This enables the classifier to capture complex patterns and contextual information inherent in the images. By developing a deep learning-based z-disc classifier, we aim to achieve a robust and more accurate classification across cells of varying maturity levels, enhancing the overall quality of sarcomere detection and subsequent analyses.

**3.2.1. Data preparation and manual labeling.** To develop a deep learning-based classifier for z-disc detection, we began with a subset of the Allen Institute dataset, designated as the "training set" in the original publication [32]. This subset comprises approximately 5,000 single-cell images acquired from the alpha-actinin-2-mEGFP channel. Using the SarcGraph z-disc segmentation framework, we first identified contours corresponding to potential z-discs and other structural elements. As illustrated in Fig 2 panel A, each detected contour is then localized in the raw cell image and a $128 \times 128$ pixel region is cropped to center that contour. This contour-centered crop serves as a single data point for our classification pipeline. To prepare the input data for our deep learning framework, we transform each processing contour's cropped region into a three-channel input image using two distinct preprocessing approaches: Type 1 preserves the original pixel intensities, while Type 2 removes pixel intensity dependence to emphasize structural features.

The first method (Type 1), referred to as "Pixel Intensity Preserving" and illustrated in Fig 2 panel A, aims to provide contextual information about the processed contour and its surrounding structures, while preserving the pixel intensity values of the raw image. Here, the three channel input consists of: (1) the cropped raw image centered on the processing contour, (2) the cropped image masked by the processing contour, and (3) the cropped image masked by all other contours except the target.

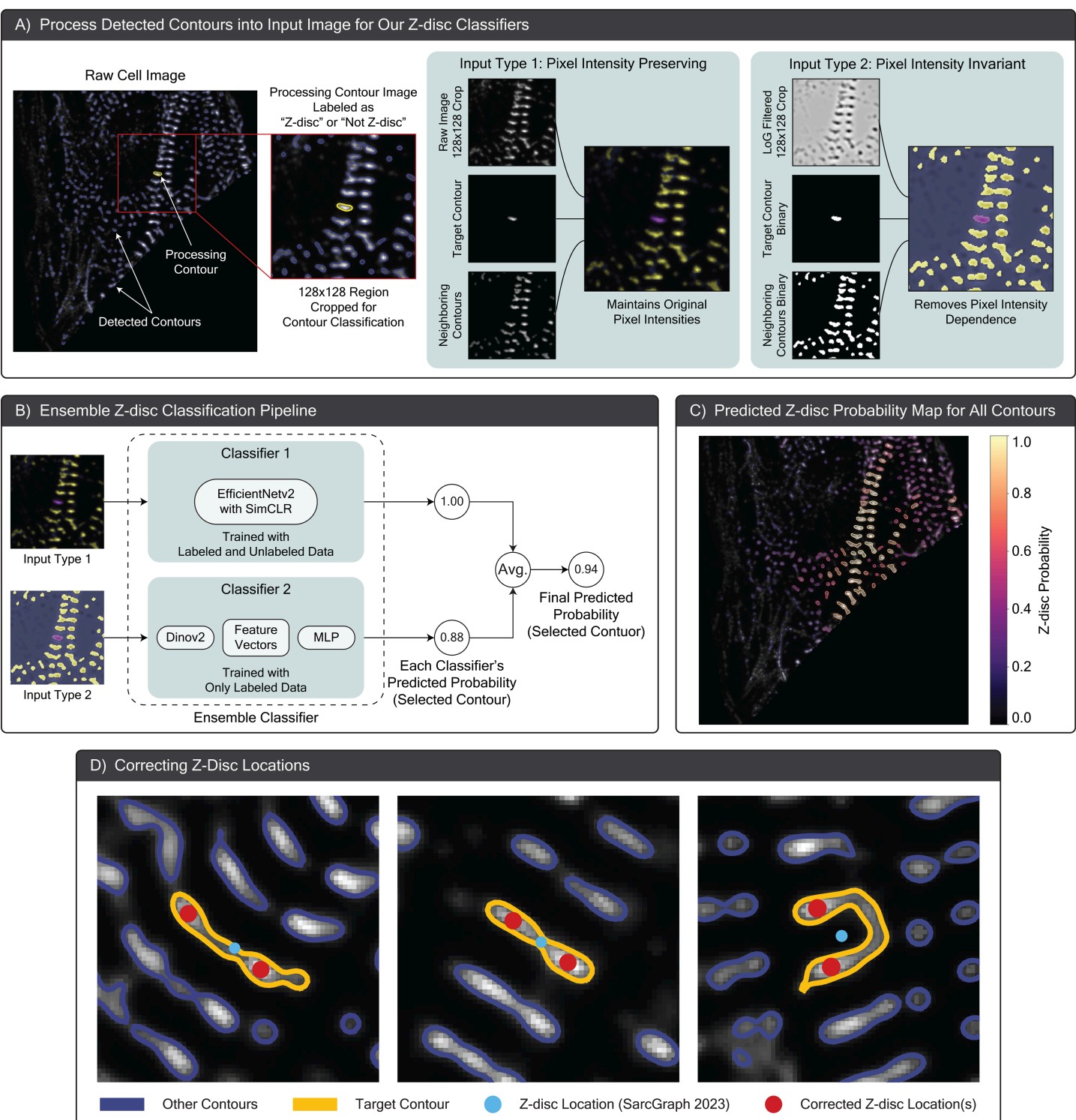

**Fig 2. Enhancements to SarcGraph's z-disc segmentation pipeline.** (A) Starting from a raw cell image with all detected contours shown in blue, each contour is processed individually to crop a $128 \times 128$ pixel region centered on it (current processing contour highlighted in yellow). This region serves as the image representation of the processing contour and is transformed into two distinct input types: Type 1, which preserves the original pixel intensities, and Type 2, which removes pixel intensity dependence. (B) depicts our ensemble classification architecture, where the two input types are processed through separate classifiers: a SimCLR-based EfficientNetv2 trained on both labeled and unlabeled data (Classifier 1), and a DINO-v2-based feature extractor followed by an MLP trained only on labeled data (Classifier 2). The final Z-disc probability is computed as the average of both classifiers' predictions. (C) shows the predicted Z-disc probabilities for all detected contours in the sample cell, where each contour is colored according to its predicted probability of being a Z-disc (higher values in warmer colors indicate higher probability of being a Z-disc).

(D) illustrates the application of our z-disc location correction method to three contours selected from different samples. The yellow line highlights the processing contour, while blue lines outline all detected neighboring contours. The blue dot marks the original z-disc location marked by the original SarcGraph, and the red dots indicate the corrected positions. Note that this approach can both refine z-disc locations and identify multiple z-discs that were previously merged into a single contour; see Sect E and Fig A8 in S1 Appendix.

The second method (Type 2), referred to as "Pixel Intensity Invariant" and illustrated in Fig 2 panel A, has less sensitivity to pixel intensity variations than the first method. Specifically, the second method was selected to focus on structural features. In this approach, the channels include: (1) the Laplacian of Gaussian (LoG)-filtered [42,53] version of the cropped raw image, (2) the binary mask of the processing contour, and (3) the binary mask of all other contours excluding the target.

Through this process, we generated approximately 2.5 million $128 \times 128$ images. For ground truth labels, we developed a graphical user interface (GUI) to manually annotate approximately 6,000 of these contour-centered images as either "z-disc" or "not z-disc." This manual annotation process provides the essential training and evaluation dataset for our deep learning model. Leveraging these labels, the classifier learns to predict a probability indicating how likely a given input image corresponds to a true z-disc, thus enabling an automated, quantitative analysis of z-disc structures.

**3.2.2. Semi-supervised training of a Z-disc classifier.** Given over 2.5 million unlabeled contour images and a manually labeled set of 6,000 contour images, semi-supervised learning [54] is particularly well-suited to leverage the full potential of this large, partially annotated dataset. Semi-supervised learning has recently emerged as an effective strategy for image classification tasks [55], which often involves an initial step of unsupervised representation learning from unlabeled images followed by a supervised fine-tuning phase using labeled images. Despite the success of this approach on famous benchmark datasets such as ImageNet [56], training these models on large sets of unlabeled images is challenging as it requires substantial computational resources and extensive hyperparameter tuning.

Considering these constraints, we developed an ensemble classification pipeline, illustrated in Fig 2 panel B. We first trained an EfficientNetV2 [57] using the SimCLR framework [58], as depicted in Fig 2 panel B (Classifier 1), to capture dataset-specific features. In this setup, unlabeled contour images were utilized to learn meaningful representations for the intensity-preserving input type, while the labeled subset was used for fine-tuning. Although this semi-supervised approach allowed our model to develop a strong, dataset-specific understanding of z-disc structures, it demanded substantial computational resources, reflecting the core challenges of training a semisupervised model from scratch. Details of the model training process are provided in Sect A in S1 Appendix.

Together with this, we used a pretrained DINOv2 [59] feature extractor—shown in Fig 2 panel B (Classifier 2)—to generate vector representations of contour images. This vision transformer-based model, was pretrained on over 140 million unlabeled images, offering robust, high-level features without the need for additional fine-tuning. We utilized the representations generated for the set of 6,000 labeled contour images by the pretrained DINOv2 model as inputs to a multilayer perceptron (MLP) trained to classify contours. This approach significantly reduced computational costs and complexity compared to our SimCLR-based model. Ultimately, we combined the predictions from both classifiers by averaging their output probabilities, leveraging the complementary strengths of each model to achieve a more robust and accurate z-disc detection system. In doing so, we leverage the established benefit of ensemble learning in improving classification accuracy and reliability [60], ultimately enhancing the z-disc detection capabilities of our pipeline.

### 3.3. Enhancements to the SarcGraph Z-disc detection pipeline

The original SarcGraph framework is not well equipped to handle immature cell with poorly developed sarcomeres. As discussed in the previous section, one major limitation is the frequent misclassification of puncta and other structures as z-discs, leading to false z-disc identification. To address this challenge, we introduced two key improvements: a z-disc classifier to enhance detection accuracy and a method to correct z-disc placement. These enhancements are detailed here.

#### 3.3.1. Integration of a Z-disc classifier in the SarcGraph Z-disc detection pipeline.

In the original SarcGraph algorithm, contours detected by global Otsu thresholding were classified as valid z-discs if their lengths fell within a predefined threshold. This approach is ineffective in less mature cells, where many non-z-disc structures (such as z-bodies or fibrous formations) could pass the length-based filtering. To address this, we integrated our trained ensemble z-disc classifier into the pipeline. For each detected contour, the classifier predicted a probability of the contour being a z-disc. We then applied a probability threshold of 0.3, filtering out contours with very low z-disc probabilities, and improving upon SarcGraph's length-based contour filtering mechanism. This modification significantly improved the precision of z-disc segmentation, especially in less organized, immature cells, where false positives were previously more frequent (e.g. see Fig 2 panel C). In addition, Fig 3 panel B-2, illustrates how the z-disc classifier effectively filters out contours with very low probabilities of being z-discs in a representative cell. As a result, only a small subset of contours with high z-disc probabilities remains for sarcomere detection. This example highlights the effectiveness of the classifier in addressing the challenge of false positives, particularly in less organized, immature cells.

#### 3.3.2. Correcting the location of detected Z-discs.

In the original SarcGraph algorithm, z-discs $(x,y)$ coordinates are placed at the geometric centroid of contour vertices, which often misrepresent the actual location of the z-disc, especially in cases where multiple z-discs are merged into a single contour, see Fig 2 panel D. Upon reviewing sample contours, we found that a better method for specifying z-disc centers was to select the region of highest pixel intensity within their respective contours. To exploit this observation, we approximated the pixel intensity as a smooth, continuous function and used a gradient-based optimization algorithm with multiple random starts to identify local intensity peaks as z-disc centers. When multiple peaks were present, secondary peaks were considered potential z-discs if their intensity was at least 60% of that of the primary peak. The probability of these secondary z-discs was adjusted based on the ratio of their intensity to the primary peak (maximum pixel intensity). Analyzing this approach on approximately 1,000 cells (176,674 individual contours), the new algorithm identified multiple z-discs within a single contour in 7% of the cases—something not possible in the original SarcGraph pipeline. Furthermore, for the remaining contours, the corrected z-disc location shifted by more than 2 pixels (average sarcomere length is 13 pixels) in approximately 7.4% of the samples. Overall, this means the new algorithm significantly altered the z-disc localization process for 13.5% of the contours. While this modification is relatively minor compared to the z-disc classifier introduced in the previous section, it complements it by improving the overall quality of z-disc detection in the pipeline, as shown in Fig 3 panel B-3.

### 3.4. Enhancements to the SarcGraph Sarcomere detection pipeline

In addition to the enhancements in the z-disc detection pipeline presented in the previous section, we improved the subsequent sarcomere detection step by introducing an ensemble graph edge scoring method and a post-processing step termed myofibril extension (see Fig 3

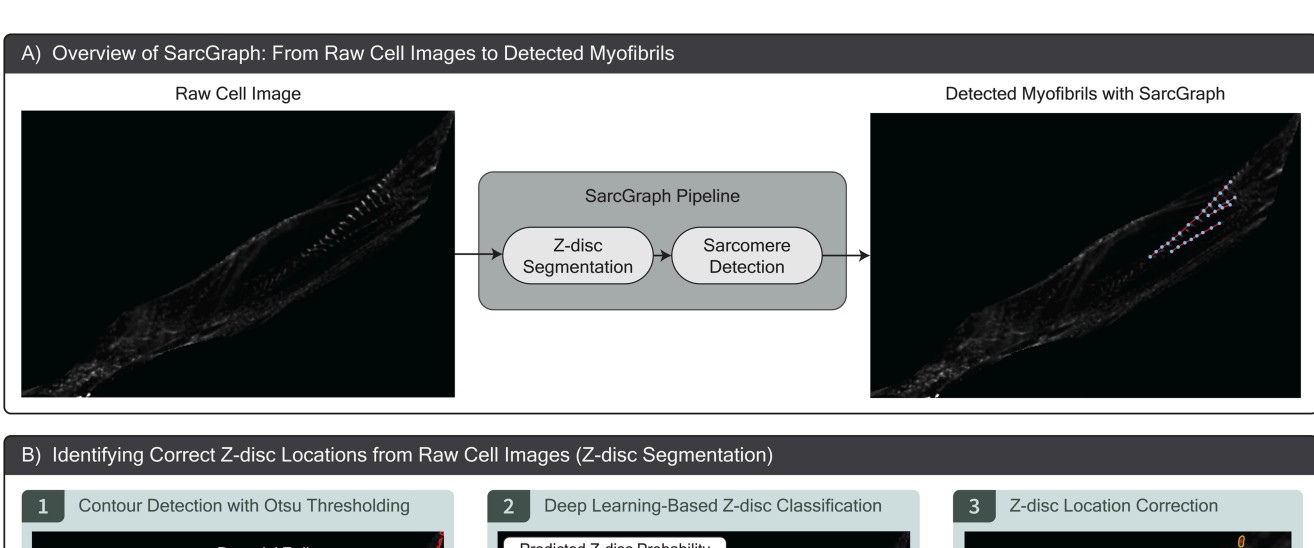

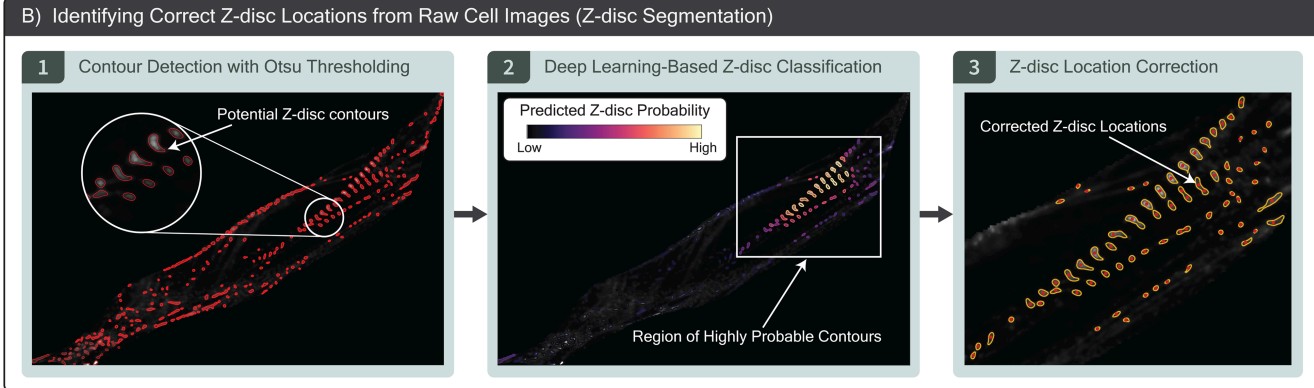

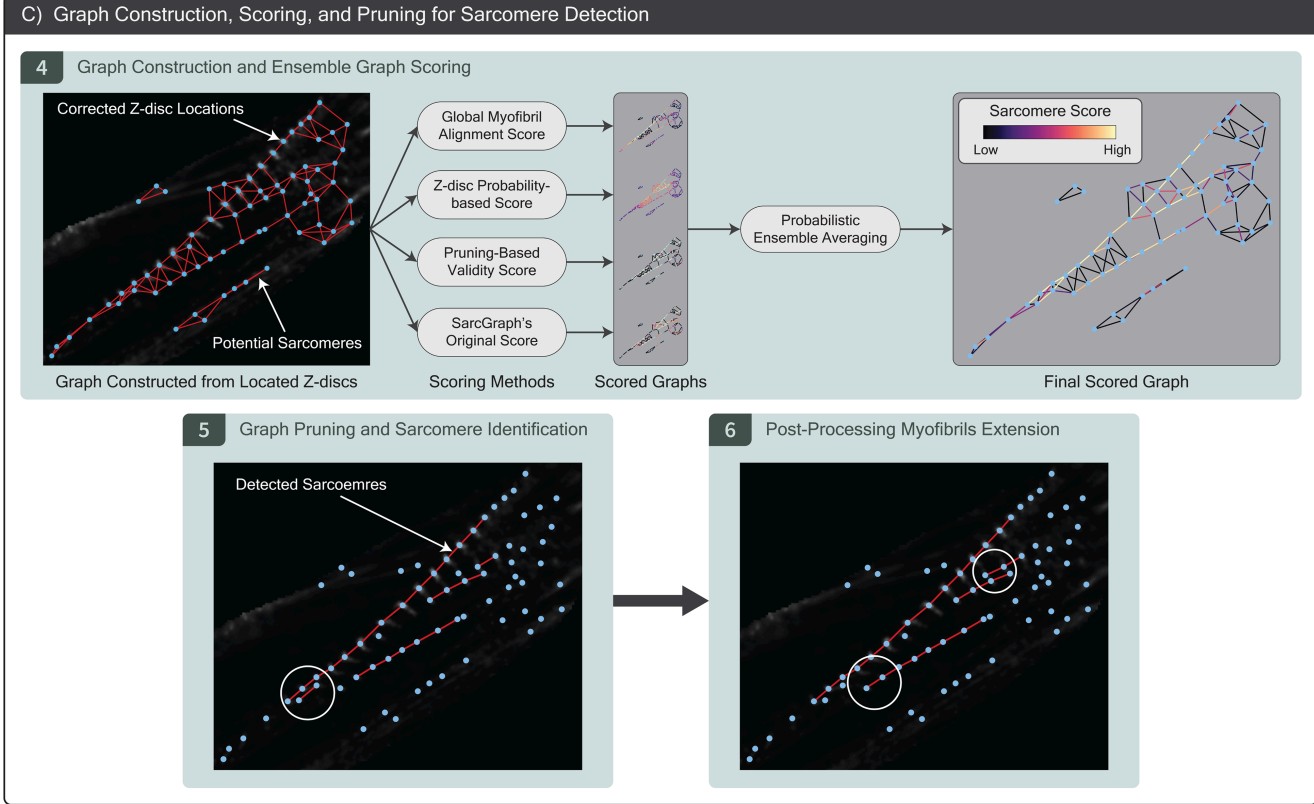

**Fig 3. The modified SarcGraph pipeline.** (A) Schematically illustrates SarcGraph taking a raw image of a cell as input and, in two steps—z-disc segmentation and sarcomere detection—outputs a list of detected sarcomeres, visualized as red lines with light blue dots indicating z-discs. (B) Visualizes the modified z-disc segmentation step in SarcGraph, where (1) shows the original Otsu-thresholding-based contour detection, (2) demonstrates the

integration of our deep-learning-based z-disc classifier (Sect 3.3.1), and (3) depicts the effect of the procedural approach used to correct the location of detected z-discs (Sect 3.3.2). (C) Illustrates the modified sarcomere detection phase in SarcGraph, where (4) schematically shows the construction of a graph by connecting detected z-discs to their *N* nearest neighbors, followed by ensemble graph scoring with probabilistic ensemble averaging (Sect 3.4.1), (5) presents the results of graph pruning, and (6) highlights the effect of applying post-processing myofibril extension to obtain the final list of detected sarcomeres (Sect 3.4.2); see Sect F and Fig A9 in S1 Appendix.

panel C). In the original SarcGraph framework, the edge scoring method used for identifying sarcomeres—where z-discs are nodes in a spatial graph and sarcomeres are edges—relied solely on evaluating immediate neighboring edges. This limitation of SarcGraph frequently resulted in the detection of short myofibrils (see Fig 4 panel B). Specifically, by selecting the edge that best matches based only on its immediate neighbors, the framework may overlook edges that, despite having a lower local score, could contribute to the identification of a longer, coherent myofibril.

To address this limitation, we introduce an updated framework that includes an ensemble graph scoring method. This approach integrates multiple scoring methods to provide a more comprehensive and robust evaluation of potential sarcomeres, thereby improving the global structural integrity of the detected myofibrils. Details of these key modifications, including the ensemble graph scoring and the myofibril extension step, are provided here.

**3.4.1. Ensemble graph scoring for Sarcomere detection.** To enhance the accuracy and robustness of the sarcomere detection process, we incorporated three additional scoring methods to complement the original edge scoring algorithm utilized by SarcGraph. Collectively, these four edge scoring methods are described as follows:

1. **Original SarcGraph Scoring:** This scoring method represents the original edge scoring algorithm implemented in SarcGraph. Each edge is evaluated based on its length, the deviation in length relative to its neighboring edges, and the angle it forms with these neighbors. Specifically, the score for edge *i* is computed as:

$$S_i = \mathbb{1}_{l_{min} < l_i < l_{max}} \left( \max_{j \in \{1,\dots,n_i\}} \left( c_1 \times f_1(\theta_{i,j}) + c_2 \times f_2(l_i, l_j) \right) + c_3 \times f_3(l_i, l_{avg}) \right) \quad (1)$$

Here, $n_i$ denotes the number of edges connected to edge *i*, $\theta_{i,j}$ represents the angle between edge *i* and its neighbor *j*, and $l_{min}$ and $l_{max}$ define the acceptable range of sarcomere lengths. The variables $l_i$ and $l_{avg}$ correspond to the length of edge *i* and the predetermined average sarcomere length, respectively. The functions $f_1, f_2$, and $f_3$ are customizable functions that yield values between 0 and 1, defined as:

$$f_1(\theta_{i,j}) = \mathbb{1}_{\theta_{i,j} \leq \pi/2} \left( 1 - \theta_{i,j}/(\pi/2) \right)^2,$$

$$f_2(l_i, l_j) = \left( 1 + \frac{|l_j - l_i|}{l_i} \right)^{-1},$$

$$f_3(l_i) = e^{-\pi \left( 1 - l_i/l_{avg} \right)^2}.$$

The coefficients $c_1, c_2$, and $c_3$ are weighting coefficients that sum to 1. By default, these coefficients are set to 1/3, allowing users to adjust them based on specific requirements (see [31] for further details).

2. **Pruning-Based Validity Scoring**: This method leverages validity values assigned to edges during the pruning step of the original SarcGraph algorithm. As edges undergo

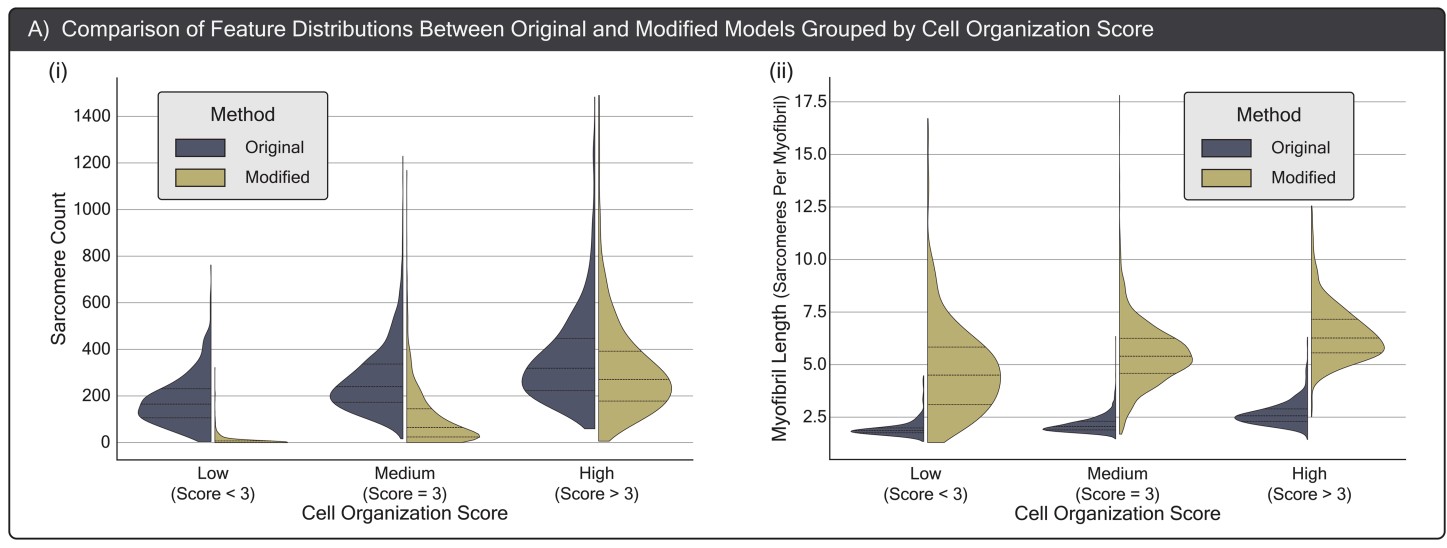

Fig 4. Here we compare the performance of the original and modified SarcGraph software for detecting sarcomeres. (A) Violin plots comparing sarcomere count and myofibril length (sarcomeres per myofibril) across samples from the Train set, grouped by cell organization score (low: score < 3, medium: score = 3, high: score > 3). With the modified version of SarcGraph, low-score cells have near zero sarcomere count. Notably, across all score categories, average myofibril length is higher with the new SarcGraph. (B) Visual comparison of sarcomere detection in one representative cell from each expert score group (1–5), showing original SarcGraph (top) versus modified SarcGraph (bottom). In low-score cells, the original pipeline yields numerous false-positive sarcomeres, whereas the modified version correctly suppresses most of these spurious detections, producing near-zero counts. In medium- and high-score cells, the modified pipeline reveals longer, more continuous myofibril chains without altering the overall sarcomere count substantially. Note that while the modified SarcGraph pipeline significantly reduces false positives and improves detection, false positives and false negatives can still occur, indicating the model still has room for potential improvement.

pruning, each edge is categorized as either rejected, weakly valid, or strongly valid, corresponding to validity scores of 0, 1, or 2, respectively. By incorporating these validity scores into the ensemble, this method ensures that greater emphasis is placed on edges that were identified as sarcomeres during the original pruning process.

3. **Z-Disc Probability-Based Scoring**: In this method, each edge is scored based on the average probability associated with its two nodes (z-discs), as determined by the z-disc classification model. This approach assigns higher weights to edges that connect two nodes with high z-disc probabilities. By incorporating this scoring mechanism into the ensemble, we aim to more directly integrate the z-disc classifier's predictions into the sarcomere detection process.

4. **Global Myofibril Alignment Scoring**: This method assigns scores to edges based on their potential to form part of a longer, continuous myofibril. The score for each edge is determined by "communicating" information between neighboring edges, where both the left and right sides of the edge (relative to its position in the graph) are evaluated for their ability to extend into a coherent myofibril. This communication process is iterated across neighboring edges, updating the score iteratively based on length and angular alignment. A detailed explanation of this scoring method can be found in Sect B in S1 Appendix.

Using these methods, we create four different scored versions of the graph and combine the results using probabilistic ensemble averaging, a technique first introduced in [61]. Probabilistic ensemble averaging is an alternative to some basic fusion methods in object detection. The key advantage of this approach over simple averaging is that when multiple methods provide high confidence scores for the same object, the ensembled confidence increases with each additional high-confidence detection. Conversely, when low-confidence scores dominate, the ensembled confidence decreases accordingly.

To compute the ensemble averaged score for an edge, let $s_i \in [0, 1]$ represent the score from the $i$-th scoring method for that edge, where $i = 1, 2, \ldots, n$ for a total of $n$ scoring methods, and let $\pi_0 \in [0, 1]$ be the prior likelihood of an edge being a sarcomere. According to the probabilistic ensemble averaging method, the posterior likelihood that an edge is a sarcomere, given the scores from all $n$ methods, is computed as:

$$p(\text{Edge} = \text{Sarcomere} \mid s_1, s_2, \ldots, s_n) \propto \frac{\prod_{i=1}^{n} s_i}{\pi_0^{n-1}} \qquad (2)$$

To normalize this, we also compute the complementary probability that an edge is not a sarcomere:

$$p(\text{Edge} = \text{Not Sarcomere} \mid s_1, s_2, \ldots, s_n) \propto \frac{\prod_{i=1}^{n} (1 - s_i)}{(1 - \pi_0)^{n-1}} \qquad (3)$$

In this work, we set $\pi_0 = 0.5$, indicating no prior preference for sarcomere or non-sarcomere classification. Therefore, the final averaged probability that an edge is a sarcomere is given by normalizing Eq 2 using Eq 3:

$$p(\text{Edge} = \text{Sarcomere} \mid s_1, s_2, \ldots, s_n) = \frac{\prod_{i=1}^{n} s_i}{\prod_{i=1}^{n} s_i + \prod_{i=1}^{n} (1 - s_i)} \qquad (4)$$

This approach combines the scores from all $n$ methods into a single probability value, resulting in an ensemble scored graph. We then apply the original SarcGraph pruning strategy

to this ensemble graph to retain only the edges that are most likely to represent valid sarcomeres. Fig 3 panel C-4, illustrates the application of the ensemble graph scoring approach on a representative sample cell, providing an overview of the process.

**3.4.2. Post-processing myofibril extension.** After detecting sarcomeres, we introduce a final post-processing step referred to as myofibril extension. This step addresses two scenarios that lead to sarcomere misdetection: first, when two myofibrils cross, SarcGraph may detect their overlapping z-disc only once, and due to how the sarcomere detection pipeline is designed, this z-disc can only be part of one of the myofibrils, creating a break in the other. Second, when a z-disc is either missed during detection or detected with a slight positional offset, it creates a break in what should be a single continuous myofibril. Both scenarios result in fragmented myofibril detection, leading to higher myofibril counts and lower average myofibril lengths.

In this step, we extend both ends of each detected myofibril by adding virtual z-discs. Each virtual z-disc is placed by copying the length and angle of the last sarcomere at each end, which determines its position. We then check whether this virtual z-disc aligns closely with either a high-probability z-disc (classified as a z-disc with higher than 0.5 probability) or another extended virtual z-disc from a nearby myofibril. To add a virtual z-disc to the myofibril, the virtual z-disc must be within 7 pixels of the matched virtual or real z-disc, the length of the newly formed sarcomere must be between 10 and 20 pixels, and the angle between the new sarcomere and the last sarcomere of the myofibril must be less than 22.5° (90°/4). If the extension satisfies all rules, we add the z-disc to the graph as a new node and include the corresponding edge as a valid sarcomere. This process can be iterated multiple times to further extend myofibrils. Finally, we remove any myofibrils that contain only a single sarcomere, retaining only those with two or more sarcomeres. Applying this post-processing step to approximately 1,000 cells increased the average number of detected sarcomeres per cell from 96.4 to 121.4, the number of myofibrils from 18.6 to 20.3, and the average myofibril length from 4.94 to 5.64 sarcomeres, demonstrating the effectiveness of the approach in recovering previously disconnected structures.

## 3.5. Machine learning for cell structural organization prediction

After performing sarcomere detection on all cell images using SarcGraph, we process the sarcomere information to obtain a set of structural features for each cell, which are then used to train a supervised machine learning model to predict cell structural organization scores that replicate expert evaluation. In this Section, we describe the structural features that serve as the input vector to our ML model, detail the machine learning approach used for score prediction, and present the metrics used to quantitatively evaluate model performance. Additionally, we explain an unsupervised clustering approach as an alternative method for quantifying cell structural organization.

**3.5.1. Cell structural feature extraction.** In this step, we explain several key features extracted from raw cell images using SarcGraph for sarcomere detection, which are then used as inputs for machine learning models to quantify cell structural organization. The features include:

- **Average Sarcomere Length**: This feature represents the average length of detected sarcomeres in a cell.
- **Standard Deviation of Sarcomere Length**: This feature captures the variation in sarcomere length within a cell. More mature cells are expected to exhibit less variation, as sarcomeres should be more uniform. In contrast, immature cells may show higher variability in this feature.

- **Orientation Order Parameter**: This feature measures the global alignment of sarcomeres within the cell [62,63]. Sarcomeres in mature cells are generally more aligned, making this a useful feature for distinguishing cell maturity.
- **Cell Area**: We define this feature as the area of the cell mask in pixels$^2$, which is provided as part of the original dataset.
- **Number of Sarcomeres**: This feature counts the total number of detected sarcomeres in a cell. A higher number of sarcomeres generally indicates better structural organization.
- **Number of Myofibrils**: This feature counts the total number of detected myofibrils in a cell. A higher number of myofibrils generally indicates better structural organization.
- **Average Number of Sarcomeres per Myofibril**: This feature is calculated by dividing the total number of sarcomeres by the number of myofibrils in the cell.
- **Z-disc Classification Ratio**: This feature represents the proportion of detected contours that are classified as z-discs (probability > 0.5) by our classifier.
- **Probabilistic Z-Disc Area**: This feature sums the product of contour area and z-disc classification probability for all contours in the cell. It represents the extent to which the cell is covered by actual z-discs, offering a continuous alternative to the Z-disc Classification Ratio.

These features are extracted from approximately 5,000 FISH cell images for training and from 1,000 FISH and 1,000 live cell images for testing. For all samples, expert-annotated cell organization scores are available and will serve as target output for our supervised machine learning models. For additional details on the feature values please refer to Sect C in S1 Appendix.

**3.5.2. Supervised machine learning model for cell structural score prediction.** To predict cell structural organization scores, we employed a Support Vector Regression (SVR) model with a radial basis function (RBF) kernel. The model hyperparameters were set to C = 10, gamma = 0.1, and epsilon = 0.5. Prior to training, we standardized all input features and normalized the number of sarcomeres, number of myofibrils, and probabilistic z-disc area by cell area, rather than using cell area as a direct input feature. We evaluated our model performance using the Pearson correlation coefficient.

**3.5.3. Unsupervised explainable clustering for cell structural score quantification.** To develop an unsupervised model for quantifying cell structural organization, we chose to apply explainable clustering. Traditional clustering algorithms, such as k-means [64], hierarchical clustering [65], and DBSCAN [66], generate clusters based on complex relationships between features through distance-based, density-based, or statistical model-based approaches. While effective, these methods produce clustering rules that are difficult to interpret. To address this limitation, recent works have focused on combining clustering with decision trees to enhance explainability. One approach involves first applying k-means clustering to partition data into $k$ clusters, and then fitting a decision tree classifier [67] with $k-1$ internal splits to approximate these clusters. However, Moshkovitz et al. [68] demonstrated that common decision tree algorithms can result in arbitrarily high cost (cost defined as the error in recreating the k-means clusters with the decision tree classifier). Instead, they proposed the Iterative Mistake Minimization algorithm, which minimizes misclassified samples at each split and guarantees the recreation cost is bounded. Frost et al. [69] further enhanced this method with ExKMC, introducing a parameter $k'$ that allows the number of leaves of the decision tree to exceed $k$. The authors have made their implementation publicly available on GitHub. In this study, we employ ExKMC to cluster the training dataset samples into three groups: "Low Organization," "Medium Organization," and "High Organization." The resulting decision tree is then used to classify samples in the test sets. We present the results of this analysis "in the Results section".

### 3.6. Notes on computational performance and scalability

We benchmarked both our z-disc classification and the full SarcGraph pipeline on representative hiPSC-CM images using an NVIDIA Tesla P100–PCIe–12GB GPU and an Intel® Xeon® E5-2680 v4 @ 2.40 GHz. On the GPU, ensemble z-disc inference requires approximately 6 seconds per cell (depending on contour count), single-model inference takes less than 1 second per cell, and the end-to-end SarcGraph pipeline completes in about 8 seconds per cell. Note that running z-disc inference on CPU-only hardware can become challenging and represent the primary computational bottleneck of the pipeline.

## 4. Results and discussion

### 4.1. Evaluating improvements in Sarcomere detection with the modified SarcGraph pipeline

Prior Sections describe enhancements made to the SarcGraph pipeline aimed at improving sarcomere detection performance, especially in cells with poorly formed sarcomeres. Here, we analyze the updated pipeline using both qualitative and quantitative methods. The two main resulting improvements are: (1) the modified pipeline detects fewer false positive sarcomeres, and (2) the modified pipeline is able to better detect long myofibrial chains.

Fig 4 panel (A-i) presents violin plots showing the distributions of Sarcomere Count (i.e., the total number of sarcomeres per cell), organized into three groups by expert-assigned organization scores. This provides a quantitative comparison of the original and modified pipelines. The sarcomere count graph shows a marked decrease in detected sarcomeres within the low score group, with most cells nearing a detected sarcomere count of 0. In contrast, cells assigned higher expert scores consistently exhibit greater sarcomere counts and there is less variation between the original and modified pipelines. Specifically, the mean sarcomere counts within each score group for the original version are 192, 262, and 342 respectively, whereas for the modified version, they are 15, 88, and 258. This highlights both the reduction in false positive sarcomere dection and the improved differentiation of structural features across score groups. To qualitatively evaluate the modified SarcGraph version, Fig 4 panel (B) shows one sample from each score tier (1 through 5), displaying the detected sarcomeres using both the original and modified SarcGraph methods. These visual comparisons, like the violin plots, highlight a decrease in false positive sarcomeres, particularly in cells with low expert score.

Fig 4 panel A-ii presents violin plots showcasing the distributions of Myofibril Length (i.e., the average count of sarcomeres per myofibril), organized into three groups by expert-assigned organization scores. The myofibril length graph indicates that in all score categories, myofibrils tend to be longer on average with the modified pipeline. Specifically, the mean myofibril lengths within each score group for the original version are 1.95, 2.15, and 2.67 sarcomeres respectively, with maximum lengths reaching 4.35, 6.27, and 6.15 sarcomeres, whereas for the modified version, they are 4.75, 5.46, and 6.48 sarcomeres, with maximum lengths reaching 16.00, 17.50, and 12.19 sarcomeres respectively. This trend of increasing myofibril length across higher score categories aligns with the expectation that cells assigned higher scores by experts exhibit greater structural organization, a feature captured better by the modified pipeline as reflected in the violin plots. In Fig 4 panel B, we also qualitatively note the longer myofibrils, particularly in cells with higher expert score.

Critically, the results shown here have two major implications. First, with this more reliable approach, Sarcomere Count and Myofibril Length can both be used as features to quantitatively describe samples. Previously, these features would capture too many false positive and

missed linkages to be sufficiently meaningful. Second, these examples showcase the efficacy of the ensemble z-disc classification pipeline introduced in the Methods section. Looking forward, we anticipate that this work could be further extended via including additional models in the ensemble, or by further tailoring the training data.

## 4.2. A closer look at the limitations of manual scoring

Within the Dataset section, we briefly touched on biases that could influence manual scoring efficacy. Here, we delve into this subject, focusing specifically on the dataset under examination. To illustrate the limitations of ordinal scoring (tiers 1 to 5), Fig 5 panel A, presents a 5x5 grid of 25 sample cells from the Train dataset. Each row corresponds to a specific expert score, increasing from 1 (top row) to 5 (bottom row). Within each row, the cells are ranked by normalized sarcomere count (normalized by cell area), with the leftmost cell having the lowest value and the rightmost the highest. Note that the horizontal axis does not represent specific numeric values, rather cells within each score group are ranked. From this array, we see that visually there is substantial variability in structural organization within individual score groups. This is observable across all scores, and is particularly pronounced in score 3, where cells with a wide range of organizational levels are grouped together likely due to the central tendency bias. Furthermore, while higher scores generally correspond to more organized cells, some cells on the far right of one row (e.g., score 2) appear more structurally organized than cells on the left of the row below them (e.g., score 3). This observation underscores the limitations of ordinal scoring, as the expert-assigned ranks do not always reflect a consistent hierarchy of structural organization across cells.

An additional measurable aspect of this dataset is the variation between scorers' ratings. Fig 5 panel B, displays confusion matrices for Expert 1 and Expert 2 across the Train, Test FISH, and Test Live datasets, with Pearson correlation coefficients of 0.679, 0.741, and 0.760, respectively which indicates strong correlation despite moderate disagreement between the experts. This analysis is inspired by Figure S2 panel J from [32], where a similar confusion matrix was plotted for a subset of 6,370 cells. Two trends are evident in both studies: Expert 1 (KG) typically assigns higher scores, while both experts frequently award mid-range scores, highlighting Central Tendency bias. Furthermore, utilizing the updated SarcGraph, we processed all cells from the train and test datasets, extracting a feature vector for each cell as outlined in the Methods section. We applied Principal Component Analysis to visualize the feature space in two dimensions and plotted the mean of the PCA-reduced feature vectors for each score group (low, medium, high) across all datasets. From this plot, shown in Fig 5 panel C, we see that while low score groups exhibit consistency across datasets in PCA-reduced feature space, medium and high score groups show a larger dispersion between the test and train sets, implying that models trained on the train dataset should be expected to predict lower scores for the sample in Test FISH and Test Live datasets relative to the expert scores. This observation mirrors a trend seen in Figure 5 panel A, from [32], where the model's predicted scores generally tend to be lower than expert scores for the test datasets. This is evident across the confusion matrices for the test sets, particularly notable for cells scored as 3 by experts but frequently predicted as 2 by the model. Consequently, any statistical model trained on these scores for prediction will encounter similar limitations.

The dataset presented in Gerbin et al. [32] is extremely comprehensive and the gold standard in the field. We note the limitations of expert scoring not to detract from the accomplishment of curating and disseminating this dataset. Rather, our goal is to add to the conversation around this work and offer new directions that can be learned from it. In the previous section, we relied on expert scores to help showcase the modifications to our pipeline. Next,

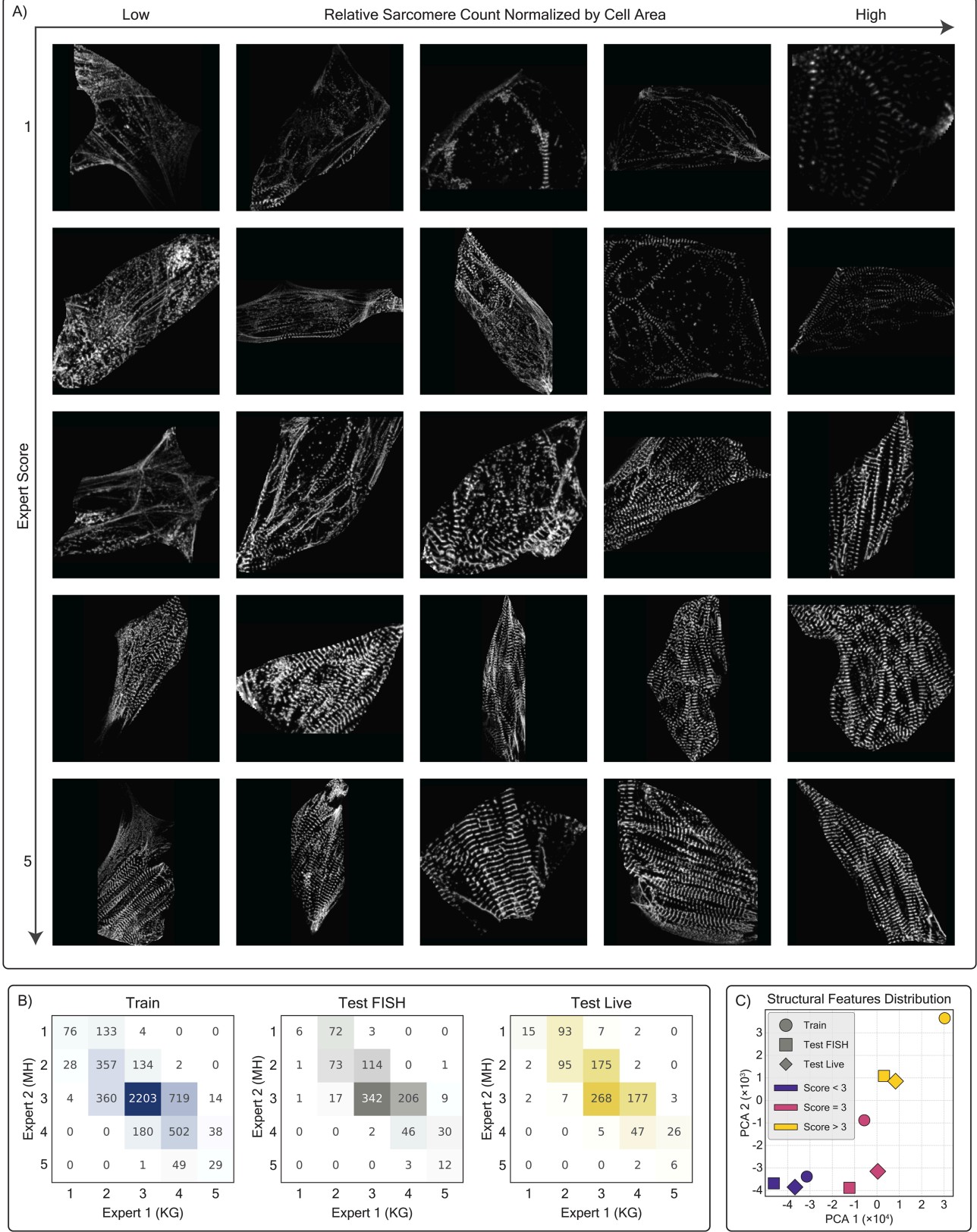

**Fig 5. This figure highlights the limitations of manual scoring.** (A) Cells arranged by sarcomere count normalized by cell area (ranked) within each score group (1 to 5), highlighting variation within score groups and visual similarity between cells from adjacent groups. (B) Confusion matrices comparing Expert 1 (KG) and Expert 2 (MH) scores across the Train, Test FISH, and Test Live datasets, emphasizing scoring differences. This panel

closely corresponds to Figure S2 panel J from [32]. (C) Average PCA projections of feature vectors from detected sarcomeres across the Train, Test FISH, and Test Live datasets, grouped by expert-assigned cell organization levels (Low: score < 3, Medium: score = 3, High: score > 3), showing distribution differences across datasets.

we will show expert score predictions based on the features that SarcGraph can extract from these data. However, it is essential to note that the efficacy of these predictions is tied to the inherent biases in expert scoring outlined in this section.

## 4.3. Machine learning-based prediction of cell structural organization score as defined by expert scores

We applied SarcGraph's sarcomere detection pipeline to extract sarcomere-related feature vectors from individual cells across the Train, Test FISH, and Test Live datasets. Using these feature vectors as inputs, listed in the Methods section, we trained a Support Vector Regression (SVR) model on the Train dataset, with the combined expert score (calculated as: expert 1 score + expert 2 score - 1) as the target. We then evaluated the model's performance by scaling the predicted scores back to the original 1-5 tier system (adding 1 and dividing by 2). Fig 6 presents the relationship between predicted scores and the average expert scores for each cell, accompanied by box plots for each score group. The model achieves Pearson correlation coefficient values of 0.85, 0.77, and 0.79 on the Train, Test FISH, and Test Live datasets, respectively. Consistent with our findings from the previous section, the model systematically predicts lower scores for samples in the Test FISH and Test Live datasets. Despite this systematic bias, our visual analysis reveals a strong correlation between predicted scores and cellular organization levels. We observed some exceptions to this trend, which can be attributed to either discrepancies in manual scoring or limitations in our image processing pipeline, where the Otsu thresholding method occasionally fails to detect z-discs that are visible to expert observers due to variations in image brightness.

To illustrate these findings, we visualized representative cells from each dataset in Fig 6, contrasting those receiving the highest predicted scores with cells showing substantial disagreement between predicted and average expert scores. While the predicted scores serve as useful indicators of cellular organization, we propose that the underlying feature vectors contain richer information about sarcomeric organization that merits deeper investigation. Rather than reducing this complexity to a single predicted score, the next section explores an alternative approach to interpreting these sarcomere-related features that moves beyond traditional manual scoring methods. For additional example visualizations and score distributions, please refer to Sect D in S1 Appendix.

## 4.4. Explainable clustering analysis for unsupervised cell organization scoring

Following our examination of supervised cell organization score prediction and its limitations in prior Sections, we explored an unsupervised approach that eliminates the need for labor-intensive and potentially biased manual labeling. Using the explainable clustering method described in the Methods section, we analyzed the Train dataset to generate a decision tree that categorizes cells into three groups: "Low Organization," "Medium Organization," and "High Organization." As shown in Fig 7 panel A-i, the decision tree uses just two key features to distinguish between all three groups: z-disc classification ratio and sarcomere density (sarcomere count normalized by cell area). Cells with no sarcomeres detected by SarcGraph are classified as low organization, while among cells with detected sarcomeres, those

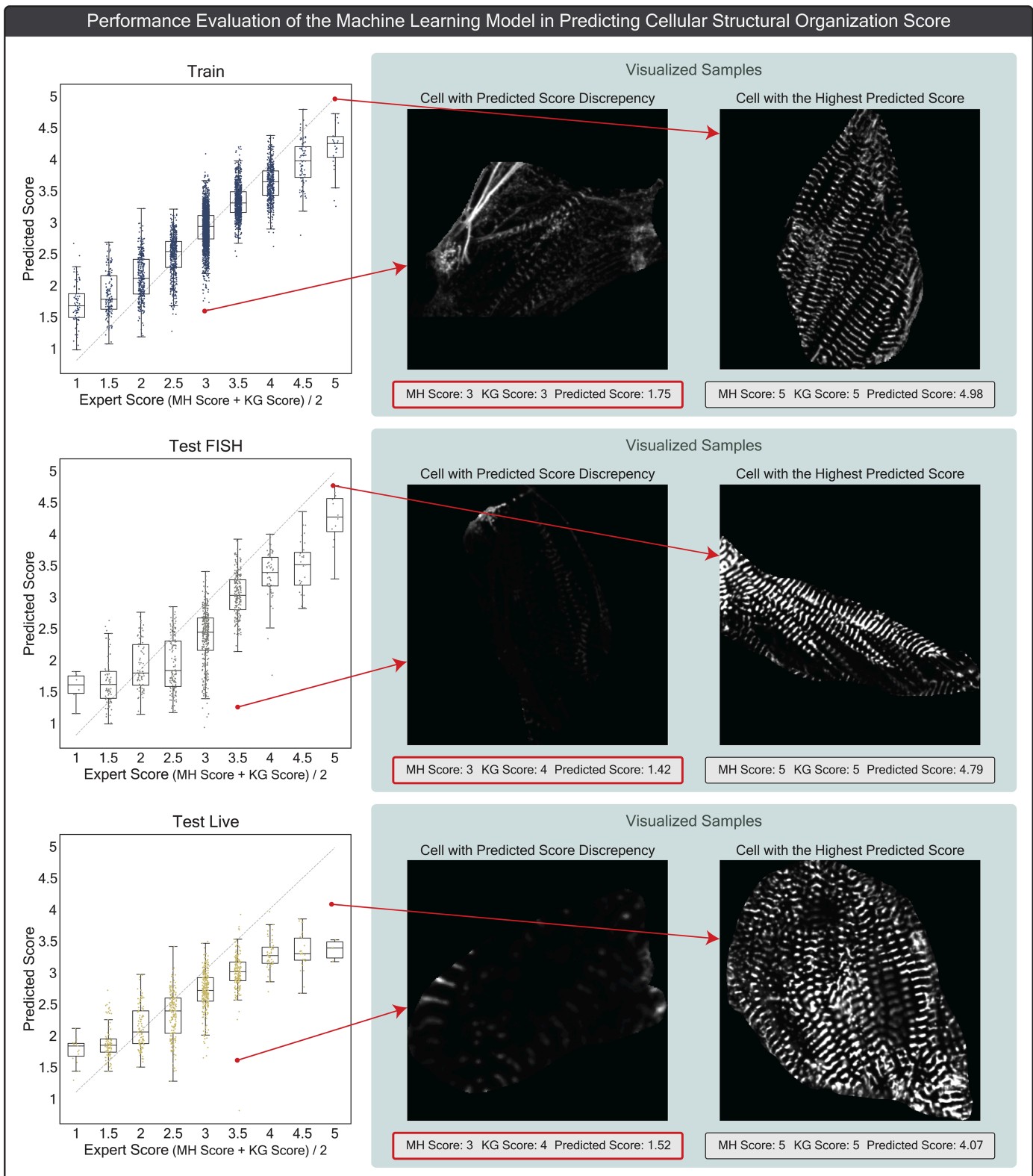

**Fig 6. This figure presents the performance of a Support Vector Regression (SVR) model on predicting cellular structural organization scores.** The model was trained on the train dataset and evaluated on two test datasets: Test FISH and Test Live. Strip plots and box plots are used to compare expert scores with model predictions across the Train, Test FISH, and Test Live datasets. The figure also highlights visual examples of cells with both the highest predicted scores and cases where there is significant discrepancy between expert and model scores.

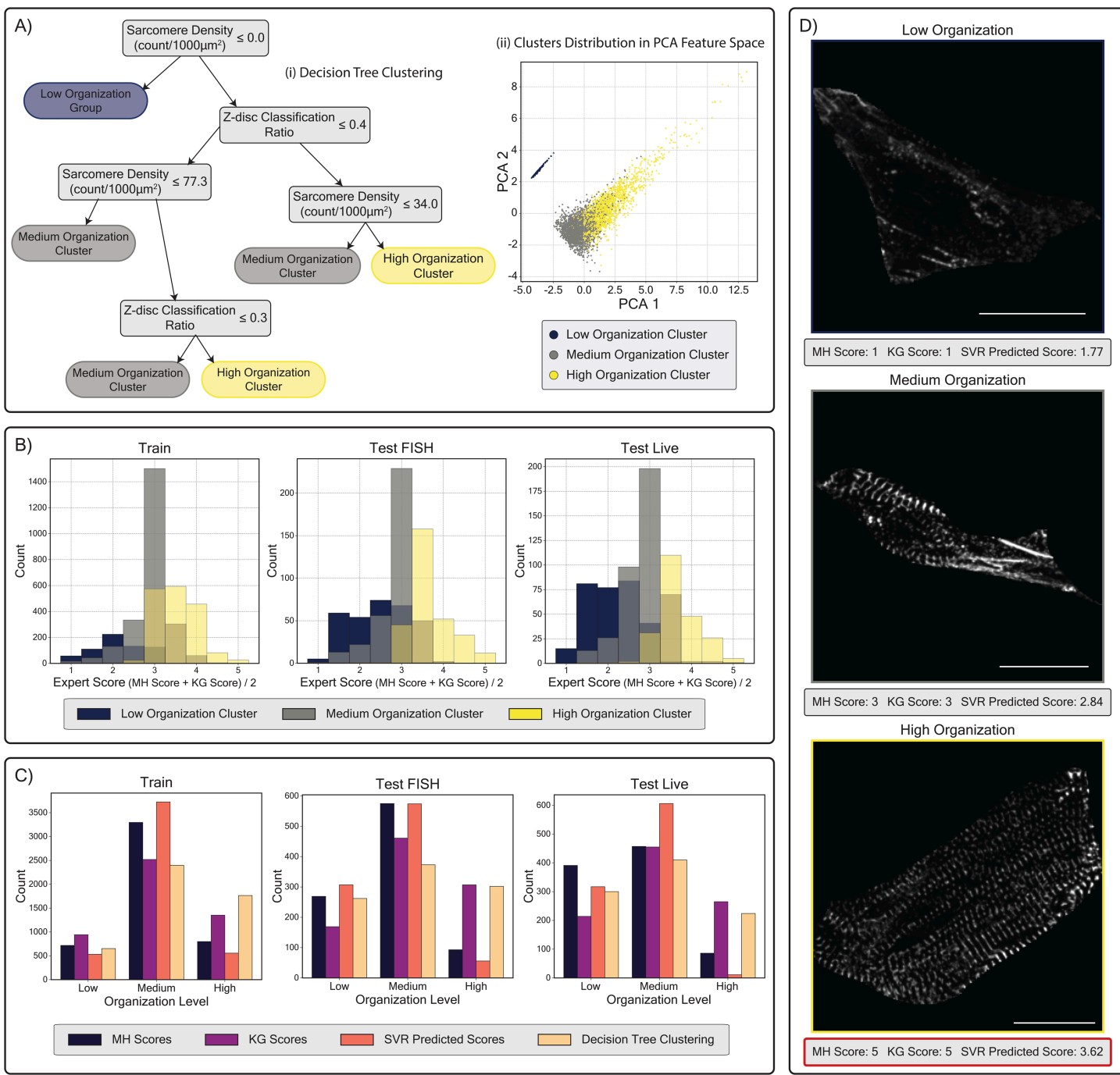

**Fig 7. Application of explainable clustering on cell features for unsupervised organization scoring.** (A-i) Decision tree derived from the Train dataset classifying cells into Low, Medium, and High organization clusters. (A-ii) Distribution of cells in 2D PCA-reduced feature space. (B) Histograms showing the distribution of average expert scores ((Expert 1 + Expert 2)/2) within each cluster for Train, Test FISH, and Test Live datasets. (C) Cell count distribution across organization levels (Low, Medium, High) for Train, Test FISH, and Test Live datasets. To compare with clustering results, expert scores are transformed into three categories (scores 1,2: Low, score 3: Medium, scores 4,5: High), and SVR predicted scores are similarly categorized (score < 2.33: Low, 2.33 ≤ score < 3.67: Medium, score ≥ 3.67: High). (D) Representative cell images from each cluster selected from the Train dataset, shown with their corresponding expert and SVR-predicted scores. Scale bars: 20 $\mu$m.

with higher z-disc classification ratio and sarcomere density are categorized as high organization, and the rest as medium organization. To examine the distribution of these clusters in the feature space, we visualized the samples in the training set using PCA-reduced dimensions (Fig 7 panel A-ii). The visualization shows complete separation of the low organization group from the other clusters, with the medium and high organization groups distributed such that samples with higher values in both PCA dimensions tend to belong to the high organization cluster.

To evaluate the relationship between our unsupervised clustering and expert scoring, we examined the distribution of average expert scores ((Expert 1 + Expert 2)/2) within each cluster across all three datasets (Fig 7 panel B). A consistent pattern emerges across the Train, Test FISH, and Test Live datasets: the low organization cluster predominantly contains cells with average expert scores between 0-3, the high organization cluster mainly comprises cells scored between 3-5, and the medium organization cluster shows the highest concentration of cells near score 3. This alignment between unsupervised clustering results and expert scoring confirms the potential of our approach, despite the imperfect agreement between our approach and manual scoring.

To quantitatively assess clustering performance, we developed a comparative framework across different scoring methods. In Fig 7 panel C, we transformed both expert and SVR-predicted scores into three categories comparable to our clustering results. For expert scores (integers 1-5), we designated scores 1-2 as low organization, 3 as medium organization, and 4-5 as high organization. For continuous SVR-predicted scores, we established thresholds at 2.33 and 3.67 to create three equally-spaced intervals. The resulting cell count distributions reveal that our clustering approach aligns well with expert scoring patterns, particularly with Expert KG's assessments. Notably, while the SVR model showed a tendency to underpredict high organization samples in both test datasets, the unsupervised clustering method maintains a more balanced distribution.

Table 1 presents the Pearson correlation coefficients between different scoring methods across all datasets. While the clustering approach shows comparable correlation with expert scores (MH and KG) and SVR predictions in the Train dataset, it demonstrates significantly higher correlation with expert scores in both Test FISH and Test Live datasets, suggesting better generalization compared to the supervised SVR model.

Furthermore, Fig 7 panel D shows representative samples from each cluster, displaying their corresponding MH scores, KG scores, and SVR-predicted scores. While these examples demonstrate ideal classification cases through our clustering method, we acknowledge the existence of misclassifications. Additional examples, including cases of misclassification, are provided in Sect D in S1 Appendix.

Together, these results demonstrate that our explainable clustering approach offers a promising alternative to supervised learning for cell organization assessment. The decision tree's interpretable rules, based on just two key features, achieve comparable or better correlation with expert scoring compared to the supervised model, particularly on test datasets.

**Table 1**. **Pearson correlation coefficients between different scoring methods: manual expert scoring (MH, KG), supervised learning predictions (SVR), and decision tree clustering (DT) across datasets.**

| | Scoring Method Pairs | | | | | |
|---|---|---|---|---|---|---|
| Dataset | MH-KG | MH-SVR | KG-SVR | MH-DT | KG-DT | SVR-DT |
| Train | 0.642 | 0.654 | 0.616 | 0.617 | 0.617 | 0.653 |
| Test FISH | 0.689 | 0.599 | 0.607 | 0.645 | 0.733 | 0.789 |
| Test Live | 0.704 | 0.621 | 0.609 | 0.697 | 0.713 | 0.746 |

While not perfect, as evidenced by some misclassifications, this unsupervised method eliminates the need for labor-intensive manual labeling while maintaining biological interpretability through clearly defined decision boundaries based on meaningful features.

## 5. Conclusion

In this work, our goal was to extend the SarcGraph computational framework to handle examples beyond the scope of the original software, and showcase the capabilities of this new version of SarcGraph on quantifying the structural organization of hiPSC-CMs. To achieve this, we utilized a publicly available dataset curated by the Allen Institute for Cell Science, which was developed for quantifying hiPSC-CM cell structural organization. In contrast with previously developed software, SarcGraph's novelty lies in its ability to detect individual sarcomeres and derive interpretable features such as the number of sarcomeres, their alignment, lengths, and other biologically relevant metrics. However, the initial version of SarcGraph encountered significant challenges, particularly in detecting z-discs and sarcomeres in less mature cells, where ambiguous structures were over-reported as z-discs. To address these limitations, we introduced two novel modifications to the pipeline. First, a deep learning-based z-disc classifier was implemented to determine whether a detected object is a true z-disc. Second, an ensemble approach was incorporated into SarcGraph's sarcomere detection algorithm, combining the previous graph-based scoring method with three additional scoring strategies to enhance detection accuracy. These modifications led to significant improvements in sarcomere detection, especially in less mature cells, while also enabling the detection of higher-quality and longer myofibrils in mature, well-organized cells. Using the improved SarcGraph pipeline, we extracted biologically meaningful features such as the total number of detected sarcomeres and the average myofibril length. We then used these features to assess cellular structural organization via two approaches: (1) a supervised learning approach, where we trained a support vector regression (SVR) model on expert-assigned organization scores from the original dataset, and (2) an unsupervised clustering approach, where we trained a decision tree to cluster cells into low, medium, and high organization groups. Notably, despite the simplicity of the unsupervised method, it achieved results comparable to the supervised approach, demonstrating its potential as a more interpretable and scalable alternative that avoids the bias of expert-assigned organization scores.

Despite these advancements, several limitations remain. SarcGraph's performance, while robust on the current dataset, has not been tested on datasets with varying imaging qualities and structural patterns, which may present significant challenges. Extending the framework to generalize across diverse datasets with minimal tuning is an important direction for future work. Additionally, the pipeline currently relies on Otsu thresholding, which is not optimal for this application. Replacing this step with modern deep learning-based segmentation methods and integrating the z-disc classification step directly into the object detection process using advanced deep learning techniques could further enhance detection accuracy and streamline the workflow. Another limitation is computational efficiency—while the original SarcGraph enabled video analysis of cell contraction, the current modified version is computationally expensive, requiring approximately 20 seconds per cell image. Future work should focus on optimizing the pipeline to enable faster analysis, especially for applications involving videos of beating cells. Additionally, in our analysis, we confirmed that with GPU acceleration the z-disc inference is not a bottleneck; however, because contour crops share much of the same image, repeated computation occurs that could be avoided by optimizing the deep learning pipeline, potentially enabling CPU-only inference. Overall, by publishing

our modified SarcGraph pipeline and trained models as open-source tools, we aim to provide researchers with an accessible and reliable framework for analyzing hiPSC-CM images. Additionally, the ensemble approach used in both z-disc detection and sarcomere detection steps in the pipeline enable future iterations of SarcGraph to seamlessly integrate new methods. We hope this work will inspire further development of novel tools and methods to more accurately analyze cellular images, advance the understanding of structural organization, and support progress in this critical area of research. Ultimately, we believe that scalable and automated frameworks like SarcGraph will revolutionize computational biology, enabling more efficient, reproducible, and impactful quantitative analyses. These advancements hold immense promise for accelerating discoveries in disease modeling, drug development, and regenerative medicine.

## Supporting information

**S1 Appendix. Extended methods, results, and comparisons.** Contains Appendices A–G, Figs A1–A12, and Tables A1–A2.
(PDF)

## Acknowledgments

We would like to thank the Research Computing Services (RCS) group at Boston University for their invaluable assistance and for providing access to the Shared Computing Cluster (SCC).

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
