## [Decision Letter · Decision Letter 0]

8 May 2025

PCOMPBIOL-D-25-00230

Quantifying hiPSC-CM structural organization at scale with deep learning-enhanced SarcGraph

PLOS Computational Biology

Dear Dr. Lejeune,

Thank you for submitting your manuscript to PLOS Computational Biology. After careful consideration, we feel that it has merit but does not fully meet PLOS Computational Biology's publication criteria as it currently stands. Therefore, we invite you to submit a revised version of the manuscript that addresses the points raised during the review process.

Please submit your revised manuscript within 60 days Jul 08 2025 11:59PM. If you will need more time than this to complete your revisions, please reply to this message or contact the journal office at ploscompbiol@plos.org. Please include the following items when submitting your revised manuscript:

We look forward to receiving your revised manuscript.

Kind regards,

Anna Grosberg, Ph.D.

Academic Editor

PLOS Computational Biology

Stacey Finley, Ph.D.

Section Editor

PLOS Computational Biology

**Additional Editor Comments :**

The reviewers' consider this work to be important to the field. However, there are important questions about the availability of all necessary components in the GitHub repository. Further, there are some methodological questions that should be addressed based on the reviewer comments.

**Journal Requirements:**

4) We notice that your supplementary Figures, and information are included in the manuscript file. Please remove them and upload them with the file type 'Supporting Information'. Please ensure that each Supporting Information file has a legend listed in the manuscript after the references list.

**Reviewers' comments:**

Reviewer's Responses to Questions

**Comments to the Authors:**

**Please note that one of the reviews is uploaded as an attachment.**

Reviewer #1: This manuscript significantly advances our ability to quantify cardiomyocyte structural organization at scale by enhancing the previously published SarcGraph pipeline. Introducing a deep-learning-based z-disc classification coupled with the ensemble graph-scoring method represents an essential methodological leap forward, particularly for challenging datasets involving immature hiPSC-CMs. Its use of graph theory and deep learning methods clearly provides strong potential for generalizability beyond classical Z-disk markers. It is highly exciting and warrants publication in PLOS Computational Biology.

The primary issue with the manuscript in its current form is the accessibility and completeness of the provided GitHub repository. The repository currently lacks crucial resources, such as the pre-trained neural network models, the labeled dataset used for training, and the clearly delineated training/testing splits. While this omission may be inadvertent or intended to avoid duplication, providing direct access to all relevant datasets and models within a single, comprehensive repository would substantially improve reproducibility and uptake by the community. Additionally, enhancing the documentation to guide new users step-by-step through replicating training and prediction would make this valuable tool even more impactful.

In addition, it would be great for the manuscript to add insights and discussions around the following:

1. Comparative Benchmarking: The original SarcGraph paper (Zhao et al., 2021) set a high standard by providing explicit numerical benchmarking against multiple state-of-the-art methods, synthetic datasets, and varied experimental conditions. It would be ideal for this new manuscript to perform similar explicit comparative benchmarking against leading tools such as SarcTrack, ZlineDetection, SOTA, SarcOptiM, and CONTRAX on identical datasets. Alternatively, the authors should expand the discussion to clarify why specific direct comparisons might not be informative or necessary, considering the different methodological goals or practical use cases between these tools.

2. Computational Scalability and Performance Metrics: Providing clear, quantitative performance metrics for computational efficiency—such as runtime benchmarks on representative large-scale datasets, including details of the computing resources used—would significantly enhance the manuscript's practical relevance, particularly given its emphasis on high-throughput analysis.

Finally, figures clarity could be improved as follows:

Figure 2D could benefit from a clearer visual representation of the z-disc correction process, perhaps by illustrating intermediate steps explicitly.

Figure 3C-4 should visually or textually differentiate between various scoring methods (pruning-based, z-disc probability-based, original SarcGraph scoring).

Figure 1A-iii is somewhat visually crowded; simplifying or enlarging the depicted contours would improve readability.

Figure 4B captions should explicitly state the specific structural differences demonstrated between the original and modified SarcGraph results.

Figures 6 and 7 could include brief annotations explaining discrepancies between predicted scores and expert scores, helping readers understand the structural features leading to these differences.

In conclusion, this manuscript represents an exciting methodological advancement with strong potential for wide-ranging applications. Addressing these highlighted issues will further enhance its clarity, rigor, and utility to the computational biology community.

Reviewer #2: This is a high quality manuscript that describes new machine-learning inspired code to analyze sarcomeres and myofibrils in microscopic images. The text is very detailed and presents strengths and limitations of the proposed algorithms. The figures are impressive. The work is likely to be of substantial value to the field.

This reviewer has only on specific comment which relates to the upload to GitHub. While the source code is (likely) all there, there is not enough information in the GitHub repo to make it easy for a new user to implement the code. Including a tutorial or some worked examples with the GitHub repo would markedly increase the impact of this work.

Reviewer #3: The review is uploaded as an attachment

**Have the authors made all data and (if applicable) computational code underlying the findings in their manuscript fully available?**

Reviewer #1: **No: **example datasets, pre-trained network, labeled datasets are missing from the repo.

Reviewer #2: Yes

Reviewer #3: Yes

PLOS authors have the option to publish the peer review history of their article (what does this mean?). If published, this will include your full peer review and any attached files.

Reviewer #1: **Yes: **Francesco Pasqualini

Reviewer #2: No

Reviewer #3: No

**Figure resubmission:**
---

## [Decision Letter · Decision Letter 1]

13 Aug 2025

Dear Dr. Lejeune,

We are pleased to inform you that your manuscript 'Quantifying hiPSC-CM structural organization at scale with deep learning-enhanced SarcGraph' has been provisionally accepted for publication in PLOS Computational Biology.

Best regards,

Anna Grosberg, Ph.D.

Academic Editor

PLOS Computational Biology

Stacey Finley, Ph.D.

Section Editor

PLOS Computational Biology

Reviewer's Responses to Questions

**Comments to the Authors:**

Reviewer #1: All comments were addressed and the qualtiy of the GitHub repo and data resources has massively improved. Thanks for your work and congrats on a great piece of research.

Reviewer #2: The authors responded appropriately to prior comments.

Reviewer #3: The authors have addressed my comments. I have no further comments.

**Have the authors made all data and (if applicable) computational code underlying the findings in their manuscript fully available?**

Reviewer #1: Yes

Reviewer #2: Yes

Reviewer #3: Yes

PLOS authors have the option to publish the peer review history of their article (what does this mean?). If published, this will include your full peer review and any attached files.

Reviewer #1: **Yes: **Francesco Pasqualini

Reviewer #2: No

Reviewer #3: No

---

## [Editor Report · Acceptance letter]

PCOMPBIOL-D-25-00230R1

Quantifying hiPSC-CM structural organization at scale with deep learning-enhanced SarcGraph

Dear Dr Lejeune,

I am pleased to inform you that your manuscript has been formally accepted for publication in PLOS Computational Biology. Your manuscript is now with our production department and you will be notified of the publication date in due course.

With kind regards,

Judit Kozma
